# The Role of Parenting, Dysregulation and Self-Esteem in Adolescents’ Problematic Social Network Site Use: A Test of Parallel and Serial Mediation Models in a Healthy Community Sample

**DOI:** 10.3390/ijerph192013154

**Published:** 2022-10-13

**Authors:** Alessandro Costantini, Cristina Semeraro, Pasquale Musso, Rosalinda Cassibba, Gabrielle Coppola

**Affiliations:** 1Department of Political Sciences, University of Bari Aldo Moro, 70121 Bari, Italy; 2Department of Education, Psychology, Communication, University of Bari Aldo Moro, 70121 Bari, Italy

**Keywords:** positive parenting, negative parenting, dysregulation profile, self-esteem, problematic social network site use

## Abstract

The study addresses some gaps in the current understanding of adolescents’ Problematic Social Network Site Use (PSNSU) by exploring the role of parenting as a precursor, and dysregulation and self-esteem as possible mediators. The sample includes 148 parents (15% fathers) and their adolescent offspring (23% male, age ranging from 14 to 18 years old, *M* = 15.96, *SD* = 1.36). Parent-reported dysregulation and positive/negative parenting style and adolescent-reported PSNSU and self-esteem were collected. As to positive parenting, simple parallel mediations were fully supported: positive parenting was associated with less dysregulation and higher self-esteem and both conditions independently predicted adolescents’ PSNSU. Additionally, a serial mediation model was confirmed, suggesting that positive parenting is associated with less PSNSU by means of the sequential effect of dysregulation on self-esteem. As to negative parenting, results only support one simple mediation: negative parenting predicted PSNSU through dysregulation. Self-esteem was not impacted by negative parenting, interrupting the indirect pathways. The direct effect of negative parenting on PSNSU was significant, suggesting a partial mediation. Findings deepen the current understanding of teens’ PSNSU and highlight the importance of targeting parenting when implementing interventions to prevent and treat PSNSU.

## 1. Introduction

According to the latest data reported by the consortium *Generazioni Connesse* (2022), supported by the Italian Ministry of Education and founded by the EU in collaboration with the Universities of Rome and Florence, 42% of the almost 2,500 surveyed Italian teens reported being online for between 5 and 10 h a day, while 12% declared being constantly online. If such results appear reassuring on one side because they show a decrease of time spent online compared to the year before (in 2021, 59% of the respondents reported being engaged online between 5 and 10 h per day and 18% declared themselves to be constantly connected), still they show a massive engagement of Italian adolescents online, which might lead to maladaptive outcomes such as poor sleep quality and school performance [1,2], besides increasing the chance of encountering risky behaviors such as cyberbullying and sexting [3,4].

These concerns have been shared some years ago by the Italian Association of Pediatrics: based on the existing evidence of the negative impact on young children of prolonged exposure to digital technology [5], Italian pediatricians recommended that caregivers correctly handle and mediate their children’s exposure to digital technology. In sum, the scientific community endorsed both putting effort into implementing educational guidelines and interventions with the aim of supporting caregivers, children and adolescents to become more aware of the many risks associated with an excessive online engagement, and pursuing a deeper understanding of the precursors of such behavior.

According to the state of art of the scientific literature, this theme appears to be quite a controversial issue and there is a variety of different labels shared by the scientific community [6]. Based on Youngs’ pioneering work [7], nowadays many scholars agree on identifying an excessive use of the internet as an addiction, as it seems to share the same features of other addictive behaviors, such as salience, frequency, duration or intensity of the behavior, withdrawal symptoms, continuation of the behavior despite family conflict, impoverished social life and frequent relapses [6]. On the other hand, a more cautious position claims that there are many different forms of problematic online engagement, each one implying a different target of the addictive behavior, whether gambling, buying, sexting or other [8], which makes it difficult to use the unique label of Internet Addiction. Moreover, Griffiths [9,10] argues that in some cases, the internet seems just a medium in support of other addictive behaviors, as for example for online gambling, shopping, cybersex and cyberporn; conversely, it is an essential part of the online activity in the case of another kind of problematic behavior, which could not take place through other means, such as social networking [9,10,11]. To say it in Griffiths and colleagues’ words [10], there is a difference between “*addictions on the Internet and addictions to the Internet*” (p. 194). Based on this argument, other scholars claim that it might be more useful to refer to internet-related disorders [10,11]. Additionally, there is evidence supporting the validity of differentiating between a Generalized Pathological Internet Use (GPIU) and a Specific Pathological Internet Use (SPIU), with the first being characterized by a more general, multi-dimensional pathological use of the Internet, while the second related to the tendency to engage in a specific function or application of the Internet, i.e., gaming or gambling [10,12,13].

In sum, evidence seems to support the existence of differentiated and multifaceted form of problematic internet related behaviors instead of a unitary problematic behavior or psychiatric disorder and therefore, there is a need now to pursue specific investigation lines to understand the peculiarities of each form as well as to implement different assessment methods for each kind of problematic behavior [10,11,13,14,15,16].

Following this line of reasoning, we will restrict our focus on adolescents’ social networking site use of Facebook, Instagram and YouTube, which is a very common behavior, but might be considered problematic in an extreme and addictive manifestation. According to the review by Kuss and Griffiths [17], social networking among teems is functional for meeting many basic needs, such as safety, self-expression, social support, information seeking and identify formation. However, when social networking use is associated with increased monitoring, compulsive checking and excessive engagement, it might be associated with maladaptation and lead to detrimental effects: albeit evidence is still limited, reviews on Problematic Social Network Site Use (PSNSU) show that this kind of behavior is linked concurrently and longitudinally to many clinical symptoms such as anxiety, stress and depression, and to signs of maladaptation such as low emotion regulation, bad sleep quality and low family functioning [18,19,20,21].

Despite this evidence linking PSNSU to psychiatric symptoms among teens, none is available on its association with the dysregulation profile, which itself has been receiving growing attention in recent years. From a developmental perspective, the achievement of effective emotional and behavioral self-regulation is an important developmental goal, and disadvantages in such a developmental path may contribute to adjustment difficulties characterized by emotional and behavioral under or over control, which can increase the risk for clinically relevant externalizing or internalizing problems in subsequent age periods (for reviews, [22,23]). The dysregulation profile is not captured by any specific disorder, but describes a self-regulatory problem in multiple domains of functioning, i.e., affective, behavioral and cognitive, and as such it is transversal to many different disorders occurring in childhood and adolescence, especially on the externalizing pole, such as ADHD and conduct disorder [24,25,26]. For this reason, it is considered nowadays as a transdiagnostic factor linked to heightened risk for psychopathology [27]. Initially captured through an index based on the Child Behavior Checklist (CBCL, [28]) scales, it has been reliably assessed also with indexes based on the Strength and Difficulties Questionnaire (SDQ, [29]) scales and items [25,26,30].

In fact, as reviewed above, the clinical symptoms linked to problematic and addictive behaviors on the internet among teens all share the common trait of dysregulation, such as ADHD [31], impulsivity traits and externalizing behaviors [32]. Indeed, such evidence has led some scholars to suggest that problematic internet use should be conceptualized as a clear sign of dysregulation, as it is characterized by high impulsivity, sensation seeking, low inhibitory control and poor decision-making abilities [33], or as maladaptive self-regulatory strategies in the absence of more functional and healthy coping strategies [34].

Restricting the focus to PSNSU, only one study has supported the direct link between dysregulation and both PSNSU and internet addiction [35]. Nevertheless, the findings refer to young adults, leaving unexplored the adolescent age which is so much at risk for both dysregulation and PSNSU. Moreover, the study conceptualized dysregulation solely as emotional, and suggests a compensatory mechanism according to which PSNSU might mitigate dysphoric mood and negative emotions. In sum, the link between teens’ dysregulation profile, as conceptualized above, and PSNSU is a gap in the current state of art of the literature.

When enlarging the focus to possible precursors of both dysregulation profile and PSNSU, quality of parenting should be considered as a good candidate. In particular, positive parenting represents a strategy by parents which facilitates involvement through effective communication and intimate interest in their child’s life, involvement with their activities, support, as well as praise, reinforcement, congratulations for the child’s achievements and physical contact. Negative parenting represents a parents’ strategy characterized by poor monitoring due to both the child not sharing his/her activities and routine and the parents’ lack of supervision of their children, together with inconsistent discipline, rule setting and punishment for misbehavior [36,37].

Although theoretical suggestions support the role of environmental influences on achieving effective regulatory skills [22,38,39,40], especially quality of caregiving, evidence on the relation between the dysregulation profile and quality of parenting is quite scant ([41] for a review) and only recently has it been tested in relation to the SDQ profile [27].

As to the relation between quality of parenting and risky engagement on the internet, many family conditions have been highlighted as possible risk factors ([42] for meta-analytic evidence): among others, insecure attachment, childhood physical and sexual abuse [43], parental authoritarian and permissive practices [44,45], poor child-parent communication [46]. Moreover, many other family risk factors such as parent-adolescent and inter-parental conflict, parents’ and siblings’ habitual alcohol use, positive parental attitudes toward adolescent smoking, alcohol, and substance use [47] increase the risk for adolescents’ problematic internet use and internet addiction.

When restricting the focus to PSNSU, the few studies available are consistent with findings regarding problematic internet use and show that adolescents’ PSNSU is predicted positively by dysfunctional and negatively by positive parenting [48,49,50]. However, evidence is still scant, and no studies have attempted to understand through which means parenting is related to PSNSU.

While negative parenting and dysregulation might favor PSNSU, self-esteem indeed might be a protective factor: findings show that adolescents’ internet and smartphone addiction are related to low self-esteem across different cultures [51,52,53,54]. As to PSNSU, the findings confirm that adolescents’ low self-esteem is associated with a more problematic use of social networks [55,56], because of the preference for online social interactions, personal negative self-image and lack of social skills. Nevertheless, this relation has never been investigated in relation to quality of parenting and dysregulation.

This study intends to fill some gaps in the existing state of art of the literature by composing in the same model many factors which have been suggested as being associated with teens’ PSNSU. Thus far, the relation between dysregulation and PSNSU at this age is unexplored, while scant evidence is available supporting the role of the quality of parenting as a possible precursor of PSNSU, and dysregulation and self-esteem as possible mediators. Based on the existing evidence, we hypothesize various indirect and direct pathways predicting teens’ PSNSU. Firstly, as to the indirect effects, both parallel and serial mediated pathways to PSNSU are expected. As to the former, we expect the quality of parenting to affect PSNSU either with the mediation of teens’ dysregulation and/or self-esteem. More specifically, we hypothesize that negative parenting increases the risk for adolescents’ dysregulation and/or low self-esteem which, in turn, increase PSNSU. Conversely, positive parenting is expected to be associated with less dysregulation and higher self-esteem, which prevents adolescents from engaging in PSNSU. As an alternative or additional pathway, we hypothesize a serial pathway predicting PSNSU, according to which negative parenting increases dysregulation, which itself enhances PSNSU because of dysregulated adolescents having lower self-esteem. Conversely, positive parenting is expected to be associated with less dysregulation, which is associated with higher self-esteem and consequently less engagement in PSNSU. As such, we expect to predict PSNSU through a serial mediation, with two mediators. Secondly, as to the direct effects, we expect parenting to be related to PSNSU, over and above the effects of the suggested mediators. The conceptual model is depicted in Figure 1.

## 2. Materials and Methods

### 2.1. Participants

The sample recruited in the present study included 148 parent–adolescent dyads. As to the parents, only one was asked to take part in the study: 15% (*n* = 22) were fathers, the remaining (*N* = 126) were mothers. They were on average 47.54 years old (*SD* = 5, range = 35–59) and had on average 12.80 (*SD* = 3.92; range = 5–18) years of education. They were all Italian and resident in two urban areas in the Apulia region in the south of Italy. Only 2% (*n* = 3) were single parents, 16% (*n* = 24) were either separated or divorced while the remaining belonged to biparental families, with both parents living together. Recruitment was carried out in six secondary schools, selected based on their central position in the urban area and therefore their ability to access a representative community sample of Southern Italian urban area adolescents. As regards participating adolescents, 23% (*n* = 34) were male, and their age ranged between 14 and 18 years old (*M* = 15.96, *SD* = 1.36). The participating schools had an equal distribution of boys and girls; nevertheless, a majority of families with girls decided to take part in the study. All were Italian, and according to school files, none were diagnosed with psychological delays/disorders. The families involved were not paid for their participation and were treated in accordance with the ethical standards outlined by the American Psychological Association and the Italian Association of Academic Psychologists (AIP, www.aipass.org). The study was approved by the Ethical Committee at the Department of Education, Psychology, Communication at the University of Bari Aldo Moro, in charge of evaluating psychological and behavioral research protocols (title of the study protocol: Generazione Z: risorse e rischi connessi all’uso di internet e dei social media. Ethics reference code: ET-21–05).

### 2.2. Procedure and Measures

The study’s purpose was explained to the participating schools’ principals who provided their informed consent. Parents were informed about the research purposes by the class coordinator, who is a teacher in charge of transmitting school communications to the families, either through a letter, an email or a WhatsApp message. Online completion was chosen due to the schools being partially in COVID-19 lockdown. Parents interested in participating were sent a link through which they provided informed consent to the participation and treatment of the data. No personal or sensitive information was requested, and to protect confidentiality, questionnaires were fully anonymous. Parents were instructed to choose an identifying code by combining the first three letters of the name and surname followed by the year of birth of their participating offspring, just for the purpose of appropriately matching parents’ and offspring’s questionnaires. Following parents’ participation, teachers provided a link only to the adolescents whose parents had expressed consent to complete the questionnaires, in an anonymous form and by using the same form of ID as their parent (first three letters of their name, first three letters of their surname followed by their year of birth). Teachers assured them that their participation was voluntary and that they could decline to participate at any time. None of the adolescents whose parents agreed to participate refused to complete the questionnaires.

#### 2.2.1. Parent-Reported Measures

*Parenting style.* The Alabama Parenting Questionnaire (APQ, [59]), adapted in Italian by Esposito, Servera, Garcia-Banda, and Del Giudice [36], was used to assess self-reported parenting style. The original version of the APQ, developed by using samples from the USA, comprises 35 items to measure five domains of parenting: parental involvement and positive parenting (positive scales), and poor monitoring/supervision, inconsistent discipline, and corporal punishment (negative scales). Seven additional items evaluating specific discipline practices other than corporal punishment are also usually included. Items are scored on a 5-point Likert-scale, ranging from 1 (never) to 5 (always), with high scores indicating adequate parenting practices for the positive scales, and inefficient parenting practices for the negative scales. Following the factor structure of the Italian version of the APQ [59], we used the following two subscales: (1) Positive Parenting (PP), including parental involvement and positive parenting subscales; (2) Negative Parenting (NP), including poor monitoring/supervision and inconsistent discipline subscales. The Cronbach’s alphas were satisfactory for both scales (0.83 for PP and 0.74 for NP).

Strength and Difficulties Questionnaire–Dysregulation Profile (SDQ–DP). Parents completed the Italian version of the Strengths and Difficulties Questionnaire (SDQ, [29]). This is a 25-item tool assessing children’s and adolescents’ emotional and behavioral difficulties. Items are scored on a 3-point Likert scale (0 = not true, 1 = somewhat true, 2 = certainly true). The SDQ includes five subscales, each consisting of five items: Prosocial Behavior, Hyperactivity–Inattention, Emotional Symptoms, Conduct Problems, and Peer Problems. Although the original version was intended to cover the age range from 4 to 17, a range of empirical evidence has proven its validity and age invariance on adolescents up to 18 years old [60,61,62,63]. Solid evidence supports the psychometric properties of the SDQ Italian version [64,65,66].

The 15-item SDQ–DP was computed by summing all the items of the Emotional Symptoms, Conduct Problems, and Hyperactivity-Inattention scales [26], and its reliability was assessed with Cronbach’s alpha, resulting in a satisfactory value of 0.64, in line with that reported by other scholars (e.g., [27]).

#### 2.2.2. Adolescents’ Reported Measures

Owning and frequency of use of a smartphone. According to the current state of art of the literature, problematic social network site use is strictly intertwined with mobile phone use [17]. Therefore, the first part of the questionnaire, besides age, sex and the currently attending school grade, asked if respondents owned a smartphone (yes/no) and how frequently they used it to navigate online, with answers ranging from 1 (never) to 4 (always).

Problematic Social Network Site Use. The Bergen Social Media Addiction Scale (BSMAS, [67]), which is an adaptation of the Bergen Facebook Addiction Scale [68], was used. In the BSMAS, the word Facebook is replaced by the phrase social media, this being defined as “Facebook, Twitter, Instagram, and the like” in the instructions to participants. The questionnaire includes six items, each assessing one of the six basic components of behavioral addiction [69]: salience, mood modification, tolerance, withdrawal, conflict, and relapse. Each item is scored on a 5-point scale ranging from 1 (very rarely) to 5 (very often). Higher scores indicate greater social media addiction. The psychometric properties of the Italian version of the BSMAS have been successfully tested by Monacis, De Palo, Griffiths and Sinatra [70]. The Cronbach’s alpha for reliability in the present study was satisfactory (0.78).

Self-esteem. The Rosenberg (1965) Self-Esteem Scale [71] (RSES, Italian validation by Prezza, Trombaccia, and Armento [72]) was used to assess adolescent’s self-esteem. It is the most widely used one-factor measure to assess self-esteem, including 10 items (e.g., “On the whole I am satisfied with myself”; five items are reverse worded) scored on a Likert-type scale ranging from 0 (strongly disagree) to 3 (strongly agree). Higher scores are indicative of higher levels of self-esteem. Reliability and validity for the RSES have been provided in several studies (for recent meta-analytic studies, see [73,74]). The Cronbach’s alpha for reliability in the present study was 0.88.

None of the measures had missing data; descriptive statistics are reported in Table 1.

## 3. Results

### 3.1. Preliminary Analyses

All analyses were conducted with the IMB SPSS package 24° Ed. Firstly, we dealt with the violation of normality in the distributions of the measures of interest (positive parenting, negative parenting and adolescents’ dysregulation profile, self-esteem and problematic internet use): skewness and kurtosis *Z* values revealed the presence of three out of five skewed values and two out of five kurtotic values (skewness/SE_skew_ and kurtosis/SE_kurt_ > 1.96), confirmed by the Shapiro–Wilk tests. Therefore, all measures were normalized through the Van der Waerden ranking procedure [75].

Secondly, descriptive statistics were run on the answers related to owning and use of a smartphone to navigate online: all but one adolescent reported to have a personal smartphone; 53% declared using it frequently to navigate online, while 43% declared doing so always. The remaining declared using the smartphone only sometimes (5 respondents) or never (1 respondent). These results suggested that these teens had free access to the internet.

Thirdly, a set of Pearson’s correlations were run to test possible associations between the variables of interest and sociodemographic features (parents’ years of education and adolescents’ age and sex). Results are reported in Table 2 and show some associations with the sociodemographic variables. Based on these findings, the main analyses were conducted controlling for parents’ years of education and adolescents’ age and sex.

### 3.2. Main Analyses

Mediation models were tested through IBM SPSS version 24 macro PROCESS [74], implementing model numbered as 6 in Hayes’ templates. According to the conceptual model depicted in Figure 1, the IV (parenting style) is modeled as affecting the DV (adolescents’ PSNSU) through four suggested pathways. Two are simple parallel mediations, each indirect pathway running from the IV to VD through one mediator (a_1_–b_1_ for M1, that is, dysregulation profile; and a_2_–b_2_ through M2, that is, self-esteem). In such cases, we are suggesting that negative parenting increases the risk for adolescents’ dysregulation and low self-esteem which, in turn, increases PSNSU. Conversely, positive parenting is associated with less dysregulation and higher self-esteem, which prevents adolescents form engaging in PSNSU. A third indirect influence passes through both M1 and M2 sequentially, with M1 affecting M2 (a_1_–d–b_2_): more specifically, we tested whether negative parenting increases dysregulation, which enhanced PSNSU because of dysregulated adolescents having lower self-esteem. Conversely, positive parenting is associated with less dysregulation, which in turn predicts higher self-esteem and consequently less PSNSU. The fourth effect of the IV is direct (c’), from the IV to the DV, without passing through either M1 or M2: as such, parenting is suggested to impact PSNSU, in a completely or partially independent way from offspring’s dysregulation and self-esteem.

In all models, parents’ years of education and adolescents’ age and sex were inserted as covariates. Moreover, given that we were dealing with data collected on fathers and mothers, the parental role (0 = father and 1 = mother) was inserted as a fourth covariate.

As displayed in Table 3, when positive parenting was the IV, both simple parallel mediations as well as serial mediation are fully supported: this means that positive parenting is associated with less dysregulation and higher self-esteem, and both conditions are independently associated with adolescents’ lower PSNSU. Additionally, the indirect effect through the two mediators is significant, suggesting that positive parenting is associated with less PSNSU by means of the sequential effects of one mediator on the next one. The impact of positive parenting on problematic internet use if fully mediated as the direct effect is non-significant.

When turning to negative parenting as the IV, results only support the simple mediation of the first but not the second mediator in the model: that is, negative parenting predicts PSNSU with the mediation of adolescents’ dysregulation. While self-esteem predicts PSNSU, it was not impacted by negative parenting, interrupting both the simple and serial mediation. Despite the significance of the indirect pathway mediated by dysregulation, the direct effect of negative parenting on problematic internet use still remains significant, suggesting a partial mediation.

## 4. Discussion

This empirical contribution aimed at building on the current state of the art of the literature concerning teens’ problematic social network site use (PSNSU) by addressing gaps related to its possible associations with their dysregulation profile and self-esteem and quality of parenting. We hypothesize various indirect (by parallel and serial mediations) and direct pathways predicting teens’ PSNSU from the quality of parenting. As to the indirect pathways, we suggested as possible mediators the teens’ dysregulation profile and self-esteem. The suggested indirect pathways were all confirmed for positive parenting and partially confirmed for negative parenting; as to the direct pathways from parenting to PSNSU, over and above the mediators, this was confirmed only for negative parenting.

Firstly, the relation of positive parenting with dysregulation and self-esteem provides support to fundamental theoretical frameworks recognizing the primary role of effective caregiving for the development of children’s well-being, regulatory functioning and positive adaptation [22,38,39,40,76]. More interestingly, we were able to enlarge the picture by proving support to a full mediation in which positive parenting predicted less PSNSU, by decreasing dysregulation and by increasing self-esteem, both independently and sequentially, with the first mediator acting on the second.

Secondly, negative parenting was related only to dysregulation, in line with recently reported evidence testing the link between parenting and dysregulation profile [27], and with a consistent corpus of findings linking dysfunctional parenting to disorders, especially externalizing ones, all characterized by dysregulation issues such as conduct disorder (CD), oppositional defiant disorder (ODD) and attention-deficit hyperactivity disorder (ADHD) ([22,39,40,77] for reviews). In line with the developmental psychopathology framework [78], adequate self-regulation requires parents’ ability to set limits for appropriate child behavior and/or misbehavior and when such parental functions fail, there is an increased risk for the child to develop dysregulation. Moreover, negative parenting was directly related to PSNSU, and indirectly through the mediation of dysregulation, suggesting that a dysfunctional parenting style might enhance the risk for PSNSU, also through a negative effect on the development of child’s regulatory skills.

Thus far, there is little evidence on the role of quality of parenting as a protective or risk factor for PSNSU: the few studies available show that adolescents’ perceived emotional bond with parents [48] and trust [49] are negatively associated with their PSNSU. Conversely, the only negative parenting features thus far related to teens’ PSNSU are separation anxiety, inhibition of exploration and individuality [48] and parental alienation [49]. Nevertheless, none of these contributions attempted to understand through which means quality of parenting might reduce or increase adolescents’ risk for PSNSU. Others have attempted to pursue such a goal, by identifying the internet-specific parenting practices which are effective in preventing teens’ excessive and problematic internet engagement. But findings from this line of investigation are quite inconsistent, as a variety of parental practices have been shown to be effective, whether this be parental communication regarding the internet (e.g., [79,80]), restrictive mediation (e.g., [81]) or instructive mediation [82]. More interestingly, none of these studies considered teens’ PSNSU but, instead, considered the more global issues of compulsive or problematic internet use only.

Given this state of art of the literature, our contribution to the existing knowledge is twofold: firstly, our findings enrich the current evidence on the relation between parenting and PSNSU, which is thus far quite limited. Secondly, our findings provide suggestions on the possible means though which parenting might act on teens’ PSNSU. In this respect, the positive parenting measure refers to effective communication and intimate interest in their child’s life, involvement with their activities, support, as well as praise, reinforcement, congratulations for the child’s achievements and physical contact. Our findings suggest that these parental features have no direct association with teens’ PSNSU when the mediators were considered, so they might not be associated with specific internet mediation practices as our results prove a full mediation. Indeed, it is by promoting more effective self-regulatory strategies and self-esteem, that is, by strengthening teens’ personal resources in terms of impulse control, executive functioning, emotion regulation, and coping with distress, that positive parenting might protect against PSNSU.

Our measure of negative parenting refers mainly to poor monitoring due to both the child not sharing his/her activities and routine as well as parents lack of supervision of their children, together with inconsistent discipline, rule setting and punishment for misbehavior. These are features consistently linked to dysregulation issues, as reviewed above. Our findings also show a direct effect of negative parenting on PSNSU, over and above the indirect effect through dysregulation: other parental effects seem to be at work, which might be internet-specific parenting practices or others to be targeted in further research.

Moving the focus from parenting to the two suggested mediators, our results show that both dysregulation and self-esteem predict in expected directions teens’ PSNSU. These findings contribute to understanding teens’ problematic social network engagement in the model of compensatory internet use [83] as well as the internet addiction account [7,84], by providing support to both.

According to the first, problematic internet use is a coping strategy: in relation to our findings, PSNSU, as predicted by low self-esteem, could be functional to cope with unpleasant and distressing experiences, such as negative self-perception, difficulties in building and maintaining social relationships and social isolation and anxiety, which might be a consequence of low self-esteem [51,53]. As such, social networks facilitate social interactions, although frequently in rapid and asynchronous ways. Moreover, participating in social networks in many cases implies setting up a personal homepage, which allows teens to experience a sense of mastery, to build a self-presentation according to desired standards and to communicate in an easy way information about themselves to others [85]. These could be effective coping strategies in the presence of low self-esteem.

Our findings relating dysregulation to PSNSU might additionally support the addiction account, according to which the use of the internet might be comparable to other forms of behavioral addiction, such as substance abuse [67]. In support of this, many deficits in self-regulation processes, both controlled aspects such as impulsivity, decision making, and inhibition of prepotent responses, and automatic ones such as sensation seeking, reward drive and punishment sensitivity, characterize both internet addiction [33] and addictive disorders [86].

Limitations of the study also need to be addressed: firstly, the associations between quality of parenting and teens’ dysregulation, self-esteem and PSNSU suffer from the use of a cross-sectional design with concurrent measures; therefore, we are unable to draw clear conclusions on the direction of the influences, as we are unable to exclude the alternative explanation which could fit our data; for instance, we might suggest that teens’ PSNSU increases their dysregulation and impacts negatively the quality of their parents’ behavior. Therefore, the investigation of parents’ determinants might benefit in the future from implementing a longitudinal design, which is the golden standard for reliably testing predictions over time. As regards the second limitation, our findings must be treated with caution as they are derived from a healthy community sample, rather than a clinical one; a deeper understanding of these relations requires the investigation of such dimensions among children diagnosed with disorders related to dysregulation. In a similar vein, our sample was recruited only in one region of Italy: thus, the generalizability of these findings requires a replication through a multi-centric approach by including samples recruited in different regions of Italy, as cultural variation across the country, for example related to parenting and children’s social behaviors, have been well established [87,88].

As to the third limitation, our study is unable to contribute to the understanding of how mothers’ and fathers’ parenting style differentially relates to the offspring’s dysregulation, self-esteem and PSNSU: due to the small sample size, we treated the parents as a whole sample, and controlled for parental role as a covariate in the main analyses. Growing evidence suggests that both mothers and fathers influence the development of offspring, and different pathways, both environmental and genetic, have been suggested [89]. The same can be said for teen’s sex: intriguingly, our sample includes a minority of boys; we are unable to state why more families with girls decided to participate compared to those with boys, although it may be because either the parent or the son was unwilling to complete the questionnaires. As a consequence, given the small number of boys, we were forced to treat teens’ sex as a covariate and we were unable to test differential pathways for boys compared to girls. Thus, future studies should address these issues and try to disentangle differential effects related to both parental roles and teens’ sex.

## 5. Conclusions

Notwithstanding these limitations, our findings add new information to the understanding of teens’ PSNSU by the use of a multi-informant design, that is, utilizing independent informants (teens’ and one parent) instead of a single informant as in many other contributions on this topic [48,49], in support of measurement validity and independence. Moreover, these findings might have important practical implications for the treatment and prevention of PSNSU. Firstly, effort should be devoted to implementing educational interventions to help teens, parents and teachers be more aware of possible risks related to a problematic social network site use; moreover, parenting seems to be a crucial target of possible preventive and treatment intervention: in this respect, a growing number of short-term programs to support parenting are being tested in the field of evidence-based preventive science [90]. These protocols to promote positive parenting have been proven to be effective in supporting both the affective and regulatory dimensions of parenting and therefore could be effective in supporting parents to prevent teens’ dysregulation and increase their well-being, thereby reducing the risk of PSNSU ([91] for a review).

## Figures and Tables

**Figure 1 ijerph-19-13154-f001:**
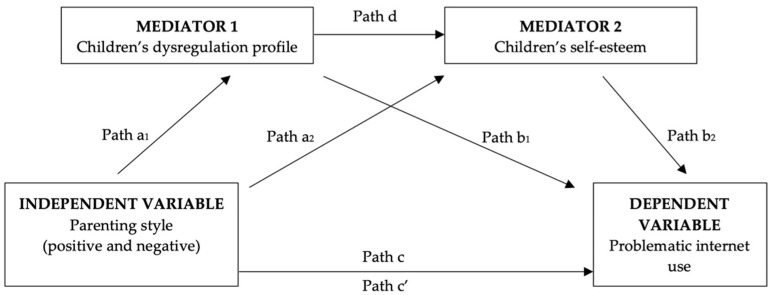
Conceptual model of the mediation linking parenting style to adolescents’ PSNSU, through parallel or sequential effects of their dysregulation and self-esteem. Note: Path c is the total effect of the IV on the DV. Path c’ is the direct effect of the IV on the outcome with the mediators in the regression, that is, after controlling for the indirect effects [57,58]. Four covariates were inserted in all models: offspring’s age and sex, parents’ role (0 = father and 1 = mother) and years of education.

**Table 1 ijerph-19-13154-t001:** Descriptive statistics (means, standard deviations and range) of the parent and adolescent reported measures.

	Mean (SD)	Range
APQ Positive Parenting	62.95 (8.46)	23–78
APQ Negative parenting	32.35 (7.86)	17–64
SDQ Dysregulation Profile	7.25 (4.95)	0–23
RSES Self Esteem	28.14 (6.38)	10–40

**Table 2 ijerph-19-13154-t002:** Bivariate correlations among the study variables.

	1.	2.	3.	4.	5.	6.	7.
1. P’s Positive Parenting							
2. P’s Negative Parenting	−0.36 ***						
3. A’s Dysregulation Profile	−0.24 **	0.34 ***					
4. A’s Problematic Internet Use	−0.21 *	0.41 ***	0.40 ***				
5. A’s Self Esteem	0.31 ***	−0.18 *	−0.46 ***	−0.40 ***			
6. A’s Sex (0 = male, 1 = female)	−0.14	0.20 *	−0.02	−0.02	−0.03		
7. A’s age (in months)	−0.03	0.12	0.22 *	0.15	−0.24 **	0.09	
8. P’s years of education	0.09	−0.11	−0.18 *	−0.10	−0.10	−0.19 *	0.03

* *p* < 0.05. ** *p* < 0.01. *** *p* < 0.001. Note. *N* is 148, there are no missing values. A stands for Adolescent, while P stands for parent.

**Table 3 ijerph-19-13154-t003:** Path coefficients of the parallel and serial mediation models predicting children’s problematic internet use (DV) from positive and negative parenting style (IV), through the child’s dysregulation profile (M1) and self-esteem (M2).

	IV	
	Parental Positive Parenting	Parental Negative Parenting
pathway	b (SE)	b (SE)
a_1_	−0.236 ** (0.079)	0.327 *** (0.078)
b_1_	0.261 ** (0.085)	0.167 * (0.084)
a_2_	0.216 ** (0.074)	−0.007 (0.080)
b_2_	−0.253 ** (0.087)	−0.268 *** (0.080)
c	−0.205 * (0.081)	0.407 *** (0.077)
c’	−0.066 (0.079)	0.312 *** (0.076)
d	−0.379 *** (0.076)	−0.430 *** (0.081)
I.E. Dysregulation profile	−0.061 (0.029)	0.057 (0.027)
C.I. [BootLLCIBootULCI]	[−0.137 −0.019]	[0.006 0.124]
I.E. of Self-Esteem	−0.055 (0.024)	0.002 (0.026)
C.I. [BootLLCIBootULCI]	[−0.113 −0.017]	[−0.054 0.053]
I.E. Dysregulation profile and Self-Esteem C.I. [BootLLCI BootULCI]	−0.023 (0.012) [−0.057 −0.006]	0.038 (0.018) [0.013 0.086]

* *p* < 0.05.** *p* < 0.01.*** *p* < 0.001. Note. Path a is the effect of the IV on the mediator; path b is the effect of the mediator on the DV. Path c is the total effect of the IV on the DV, that is, the sum of the direct and indirect effects. Path c’ is the direct effect of the IV on the outcome with the mediator in the regression, that is, after controlling for the indirect effect. In order to prove mediation, paths *a*, *b* and *c* must be significant; moreover, non-significant *b* values for path *c’* indicate a full mediation [57,58]. Parents’ role and years of education, together with adolescents’ age and sex were treated as covariates in both models. I. E. is each indirect effect, that is, the effect of the IV on the DV mediated by the mediator. C. I. is the lower-level bootstrap confidence interval (95%) and upper-level bootstrap confidence interval (95%). The number of bootstrap samples for bias corrected bootstrap confidence intervals were 20,000. Confidence intervals not containing zero mean that *b* values are statistically different from zero.

## Data Availability

The data that support the findings of this study are available on request from the first author.

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
