# Peer review of "The Role of Parenting, Dysregulation and Self-Esteem in Adolescents’ Problematic Social Network Site Use: A Test of Parallel and Serial Mediation Models in a Healthy Community Sample"

_ijerph, 2022, doi:10.3390/ijerph192013154_

Round 1

Reviewer 1 Report

The parents possible role limiting social nwtwork site use is a valuable suggestion

This  paper  deals  with  a  very  common  adolescent  social  problem .

 Its  relevance  is  obvious , although   not  very  relevant.

 Literature  at  this  regard  is  very  scanty .

Significant  data  are  reported  on  a  rather  large  number   of  families   studied

The  parents  possible   role   limiting   social  network  site  use  by  their  children   is  documented

Data  analysis  is  correct .

In  conclusion  I   suggest   to    accept     this     paper  for  publication

Author Response

We thank the reviewer for his/her appreciation and comments.

Reviewer 2 Report

The authors presented an interesting study on the role of parenting on problematic social network usage. Overall, the manuscript is clear and well written, the introduction covers relevant literature and methods and results are clearly presented.

I have a few minor concerns on the form of the report, that I have detailed below.

- The introduction discuss about Positive and Negative parenting, however no clear definition of what is Positive Parenting and what is Negative Parenting is provided, nor in the introduction nor in the methods section when the APQ survey is presented. In order to provide more context for the present research and to improve the flow of the article, I would suggest the authors to define what positive and negative parenting are.

- Some sentences in the introduction may require more referencing or details. For example, the statement “besides increasing the chance to encounter in risky behaviors, as cyber-bulling and sexting” may benefit from a reference, as as it is it's not clear if the assumption is drawn from ref 1, 2 or not. Similarly, some sentences may benefits from additional details. For example (Line 65) “internet seems just a medium in support of other addictive behaviors, while it is an essential part of the online activity in the case of other kinds of problematic behaviors, which could not take place through other means, i.e., social networking [7].” may be improved by indicating examples of other addictive behaviors.

- Figure 1. It is not clear whether the red underlining in Figure 1 are indication of specific processes or if it's an error due to a Word to Picture export. I would suggest the authors to check the figure.

- Line 278. For reproducibility purposes, please also add the operating system on which SPSS was used.

- Limitations. While the authors clearly stated the limitations of their study, I think one big limitation is missing. Data have been collected only from a specific area in Italy and as such the generalizability of the model presented may be difficult. I would suggest the authors to address this limitation in the paragraph.

Overall the manuscript is well written, scientifically sound, and may be of interest to a general reader. I would suggest the editor to consider the manuscript for publication after edits to improve the readability and reproducibility of the manuscript have been addressed.

Author Response

The authors presented an interesting study on the role of parenting on problematic social network usage. Overall, the manuscript is clear and well written, the introduction covers relevant literature and methods and results are clearly presented.

I have a few minor concerns on the form of the report, that I have detailed below.

- The introduction discuss about Positive and Negative parenting, however no clear definition of what is Positive Parenting and what is Negative Parenting is provided, nor in the introduction nor in the methods section when the APQ survey is presented. In order to provide more context for the present research and to improve the flow of the article, I would suggest the authors to define what positive and negative parenting are.

The Reviewer is right. We have added a clear definition of positive and negative parenting in the introduction, supported by adequate quotations. The added paragraph is copied and pasted following by (please note, quotations here are reported with the authors’ surnames for your fast comprehension; in the revised version of the manuscript, we have formatted using numbers, as required):

In particular, positive parenting represents a parents’ strategy which facilitates involvement through effective communication and intimate interest for their child’s lives, involvement with their activities, support, as well as praise, reinforcement, congratulations for the child’s achievements and physical contact. Negative parenting represents a parents’ strategy characterized by poor monitoring due to both the child not sharing his/her activities and routine as well as parents lacking to supervise their children, together with inconsistent discipline, rule setting and punishment for misbehavior (Esposito, Servera, Garcia-Banda, & Del Giudice, 2016; Zlomke, Bauman, & Lamport, 2015).”

- Some sentences in the introduction may require more referencing or details. For example, the statement “besides increasing the chance to encounter in risky behaviors, as cyber-bulling and sexting” may benefit from a reference, as as it is it's not clear if the assumption is drawn from ref 1, 2 or not.

The Reviewer is right. Nor ref 1 or 2 is supporting adequately the statement. We have now integrated quoting two recent reviews on the relation between problematic internet use and each of the two risky behaviors among adolescents.

Similarly, some sentences may benefits from additional details. For example (Line 65) “internet seems just a medium in support of other addictive behaviors, while it is an essential part of the online activity in the case of other kinds of problematic behaviors, which could not take place through other means, i.e., social networking [7].” may be improved by indicating examples of other addictive behaviors. 

We agree that the sentence was not clear. We now quote the authors who made this argument and explain in more depth their position, with appropriate quotations. Following by, we copied and pasted the revised sentence (please note, quotations here are reported with the authors’ surnames for your fast comprehension; in the revised version of the manuscript, we have formatted using numbers, as required):

“Moreover, Griffiths [2000; Griffiths, Kuss, Billieux, Pontes, 2016] argues that in some cases internet seems just a medium in support of other addictive behaviors, as for example for online gambling, shopping, cybersex and cyberporn; conversely, it is an essential part of the online activity in the case of another kind of problematic behavior, which could not take place through other means, such social networking [Király et al., 2016; Griffiths [2000; Griffiths, Kuss, Billieux, Pontes, 2016]. To say it in Griffiths and colleagues’ words [Griffiths, Kuss, Billieux, Pontes, 2016], there is a difference between “addictions on the Internet and addictions to the Internet” (p. 194).

- Figure 1. It is not clear whether the red underlining in Figure 1 are indication of specific processes or if it's an error due to a Word to Picture export. I would suggest the authors to check the figure. 

I apologize for this error during the picture export process. I have corrected and the red lines are no longer visible.

- Line 278. For reproducibility purposes, please also add the operating system on which SPSS was used.

I have added the SPSS version as following: “IBM SPSS version 24”.

- Limitations. While the authors clearly stated the limitations of their study, I think one big limitation is missing. Data have been collected only from a specific area in Italy and as such the generalizability of the model presented may be difficult. I would suggest the authors to address this limitation in the paragraph.

We thank the Reviewer for this suggestion. We have integrated this limitation as following (again, please note, quotations here are reported with the authors’ surnames for your fast comprehension; in the revised version of the manuscript, we have formatted using numbers, as required)::

On a similar vein, our sample was recruited only in one Region of Italy: as such, the generalizability of these findings requires a replication through a multi-centric approach, by including samples recruited in different regions of Italy, as cultural variation across the country, for example related to parenting and children’s social behaviors, have been well established [e.g., Bornstein, Cote, Venuti, 2001; Casiglia, Lo Coco, Zappulla, 1998].

Overall the manuscript is well written, scientifically sound, and may be of interest to a general reader. I would suggest the editor to consider the manuscript for publication after edits to improve the readability and reproducibility of the manuscript have been addressed.

We thank the Reviewer for his/her helpful comments.